# Investigation of the Characteristics and Antibacterial Activity of Polymer-Modified Copper Oxide Nanoparticles

**DOI:** 10.3390/ijms222312913

**Published:** 2021-11-29

**Authors:** Nan-Fu Chen, Yu-Hsiang Liao, Pei-Ying Lin, Wu-Fu Chen, Zhi-Hong Wen, Shuchen Hsieh

**Affiliations:** 1Division of Neurosurgery, Department of Surgery, Kaohsiung Armed Forces General Hospital, Kaohsiung 80284, Taiwan; chen06688@gmail.com; 2Institute of Medical Science and Technology, National Sun Yat-sen University, Kaohsiung 804201, Taiwan; 3Center for General Education, Cheng Shiu University, Kaohsiung 833301, Taiwan; 4Department of Chemistry, National Sun Yat-sen University, Kaohsiung 80424, Taiwan; roro29701@gmail.com (Y.-H.L.); phoebe00315@yahoo.com.tw (P.-Y.L.); 5Department of Neurosurgery, Kaohsiung Chang Gung Memorial Hospital and Chang Gung University College of Medicine, Kaohsiung 833301, Taiwan; ma4949@adm.cgmh.org.tw; 6Department of Neurosurgery, Xiamen Chang Gung Hospital, Xiamen 361126, China; 7Department of Marine Biotechnology and Resources, National Sun Yat-sen University, Kaohsiung 80424, Taiwan; wzh@mail.nsysu.edu.tw; 8School of Pharmacy, College of Pharmacy, Kaohsiung Medical University, Kaohsiung 80708, Taiwan; 9Regenerative Medicine and Cell Therapy Research Center, Kaohsiung Medical University, Kaohsiung 80708, Taiwan

**Keywords:** copper oxide, nanoparticles, polymer, antibacterial agents, biocompatibility

## Abstract

The proliferation of drug-resistant pathogens continues to increase, giving rise to serious public health concerns. Many researchers have formulated metal oxide nanoparticles for use as novel antibacterial agents. In the present study, copper oxide (CuO) was synthesized by simple hydrothermal synthesis, and doping was performed to introduce different polymers onto the NP surface for bacteriostasis optimization. The polymer-modified CuO NPs were analyzed further with XRD, FTIR, TEM, DLS and zeta potential to study their morphology, size, and the charge of the substrate. The results indicate that polymer-modified CuO NPs had a significantly higher bacteriostatic rate than unmodified CuO NPs. In particular, polydopamine (PDA)-modified CuO (CuO-PDA) NPs, which carry a weakly negative surface charge, exhibited excellent antibacterial effects, with a bacteriostatic rate of up to 85.8 ± 0.2% within 3 h. When compared to other polymer-modified CuO NPs, CuO-PDA NPs exhibited superior bacteriostatic activity due to their smaller size, surface charge, and favorable van der Waals interactions. This may be attributed to the fact that the CuO-PDA NPs had relatively lipophilic structures at pH 7.4, which increased their affinity for the lipopolysaccharide-containing outer membrane of the Gram-negative bacterium *Escherichia coli.*

## 1. Introduction

Advancements in medical technology have achieved new milestones, with innovative diagnostic methods and new treatment processes [1]. Public health-related issues, such as cancers, Alzheimer’s disease, cerebrovascular disease, bacterial infection, viruses, etc., which were previously considered to be incurable diseases, are now being effectively managed and controlled. However, unhealthy lifestyles and a lack of awareness in maintaining public hygiene contribute to a continuous rise in infection rates, where resulting infections may aggravate diseases and/or result in death [2]. The increasing trends in pathogen-caused infectious diseases and antibiotic resistance [3] pose a serious health concern. Therefore, the development of novel antibacterial materials has become a common goal among many researchers. Antibacterial materials can be broadly divided into the following two categories based on their chemical properties: (1) organic antibacterial materials, which are characterized by their high performance and rapid action; (2) inorganic antibacterial materials, which provide safe and long-lasting action. Although organic antibacterial materials offer the advantage of superior performance, their short duration of action and high toxicity lead to environmental pollution and adverse effects on human health [4]. In contrast, inorganic antibacterial materials possess good chemical and physical properties, produce less environmental pollution, and affect human health to a smaller extent. Currently, most studies have focused on combining nanomaterial science techniques with the inherent antibacterial activity of inorganic metal oxides [5,6], to develop metal oxide nanoparticles for use as novel antibacterial materials [7,8]. The application of metal oxides is a suitable alternative to the current antibacterial methods, as most metal oxides provide sterilizing effects. Metal oxides commonly used for antibacterial purposes include the following: Ag_2_O, ZnO, SiO_2_, CuO, MgO, and CaO [9,10]. Several studies have reported that copper oxide is highly sensitive to Gram-positive (*B. subtilis CN2*, Candida albicans) and Gram-negative (*P. aeruginosa CB1*, *E. coli*) microorganisms [11,12,13]. To enhance the performance of copper oxide (CuO)-based antibacterial materials, surface modifications are usually performed with biocompatible surface-coating materials, such as polymers [14], chitosan [15], and polydopamine (PDA) [16]. Thus, materials with different surface functional groups (for example, carboxyl groups, amine groups, and aromatic rings) [17,18,19] that can achieve higher antibacterial activity are developed. Previous studies have demonstrated the potential of CuO in antibacterial applications [20]. The three main factors contributing to the antibacterial mechanisms of CuO nanoparticles (NPs) reported in the current literature are the small size of the NPs, the release of Cu ions [21], and surface charge attraction. However, the interfacial interactions between surface functional groups with different lipophilic properties and bacteria have not yet been investigated in depth [22]. Considering that Gram-negative bacteria are enclosed by a lipopolysaccharide-containing outer membrane that serves as a barrier to antibacterial agents, the interactions of the lipophilic components of antibacterial materials with the surfaces of Gram-negative bacteria are extremely important [23]. Therefore, the investigation of the relationships between the physicochemical properties of polymer-modified metal oxide NPs and their lipophilicity and antibacterial activity towards Gram-negative bacteria, which have yet to be fully elucidated, is of great interest to many researchers [24].

In the present study, copper oxide nanoparticles were prepared using a hydrothermal method, and were subsequently subjected to modifications for surface-charge alteration through the addition of different polymers. The various nanoparticles were then analyzed to assess their antibacterial abilities and investigate their antibacterial mechanisms. The results indicated that the polymer-modified CuO NPs had a significantly high antibacterial activity. This may be attributed to the mutual electrostatic attraction between the positively charged surfaces of the modified nanoparticles and the negatively charged surfaces of *Escherichia coli* (*E. coli*), which resulted in damage and rupture of the bacterial cells. In addition to exploring the effects of particle size and surface charge, this work also aimed to determine the modification of relatively lipophilic polymers, which could significantly increase the antibacterial activity of CuO NPs towards *E. coli*.

## 2. Results and Discussion

### 2.1. Characteristics of the CuO Nanoparticles

The metal oxide species and crystal plane structures of the unmodified and modified CuO nanoparticles were identified by X-ray diffraction (XRD) (Figure 1). Diffraction peaks were present at 2θ = 32.6, 35.6, 38.8, 48.9, 58.5, and 61.7° [25,26], which corresponded to the (110), ($\bar{1}$11), (111), ($\bar{2}$02), (202), and ($\bar{1}$13) planes, respectively (indicated by black squares), and were consistent with the standard XRD powder spectrum of CuO (reference PDF card No. 65-2309). This confirmed that the synthesized nanoparticles were CuO (Figure 1a). From the X-ray diffraction peak ($\bar{1}$11) widths, the average crystallite size (D) was estimated using the Scherrer formula. The estimated crystallite sizes are 24.84, 23.30, 22.86, 24.03, and 21.79 nm for pure CuO, CuO-PEG, CuO-PVP, CuO-PDA, and CuO-PVA, respectively. The crystallite size of polymer-modified CuO is smaller than pure CuO, which can be attributed to the Cu^2+^ ions forming a complex with the polymer [27,28,29,30]. From the XRD spectra of polyethylene glycol (PEG) and polyvinylpyrrolidone (PVP)-modified CuO nanoparticles (Figure 1b,c), it can be observed that the characteristic peaks matched the diffraction peaks of pure CuO, showing that intact CuO crystal plane structures were retained in the surface-modified samples. In the XRD spectra of PDA and polyvinyl alcohol (PVA)-modified CuO nanoparticles (Figure 1d,e), in addition to the characteristic signals of CuO, two diffraction signals at 2θ = 36.5 and 42.4°, corresponding to the (111) and (200) planes, respectively, were observed (indicated by red circles), which were consistent with the standard XRD powder spectrum of Cu_2_O (reference PDF card No. 77-0199). Previous studies have shown PDA reduction of silver ions to silver atoms, due to the reducing ability of the catechol groups in PDA, which leads to electron loss [31]. As functional groups similar to catechol are present in both PDA and PVA, it was deduced that both modification agents possess reducing abilities and could result in a reduction in CuO [32]. Therefore, the addition of PDA or PVA indirectly caused a reduction of a certain amount of CuO to Cu_2_O.

Fourier-transform infrared (FTIR) spectroscopy was used to identify the functional groups on the surfaces of the various CuO nanoparticles. As shown in Figure 2, three clear signals can be observed in the spectrum of the pure CuO nanoparticles. These correspond to stretching vibrations of Cu–O in the ($\bar{2}$02) direction at 610 and 495 cm^−1^, and stretching vibrations of Cu–O in the (202) direction at 416 cm^−1^ (Figure 2a) [33,34]. As observed in the spectrum of the PEG-modified CuO nanoparticles (Figure 2b), characteristic signals for PEG were observed at 1061, 1411, and 2940 cm^−1^, which correspond to C–O vibrations, C–H bending, and CH_2_ stretching, respectively [35,36]. Figure 2c, representing the spectrum of PVP-modified CuO nanoparticles, clearly displays the characteristic signals of PVP, corresponding to C–N stretching (1019 cm^−1^), C–O vibrations (1288 cm^−1^), CH_2_ bending (1373 cm^−1^), C=O vibrations (1658 cm^−1^), and CH_2_ stretching (2940 cm^−1^). The spectrum of PDA-modified CuO nanoparticles (Figure 2d) shows signals at 1291, 1395, 1518, and 1582 cm^−1^, which correspond to the stretching and bending vibrations of C–O–H groups, amino group (N–H) shear vibrations, and C=C vibrations in the benzene rings of PDA, respectively [37]. As shown in Figure 2e, the characteristic signals of PVA-modified CuO nanoparticles are located at 1433, 1663 (vibrations of C=C), and 2940 cm^−1^ (stretching vibrations of CH_2_) [38]. The O–H signal that appeared at ≈3400 cm^−1^ in the spectra of most of the nanoparticles may have been caused by the adsorption of atmospheric water vapor onto the sample surfaces [38]. Based on these results, it was confirmed that the synthesized nanoparticles were CuO particles, and that the intended surface modifications had been successfully achieved.

The morphologies of the pure and modified CuO nanoparticles were observed by transmission electron microscopy (TEM). The lattice fringe of CuO shown in the inset image has lattice spacing of 0.27 nm (110), 0.25 nm ($\bar{1}$11), and 0.23 nm (111), respectively. The EDS analysis was obtained from the TEM data, as shown in Figure A1. The elemental analysis results of the pure CuO show that the composition consists of Cu and O elements. On the other hand, the C element could be clearly observed on different polymer-modified CuOs. Figure 3a shows the TEM image of the CuO nanoparticles prepared using hydrothermal synthesis. The nanoparticles are rectangular in shape and relatively large in size (≈834.8 nm) (Figure 3f). From the TEM image of the PEG-modified CuO nanoparticles (Figure 3b), it was observed that the modified CuO nanoparticles retained their rectangular shape, but had a smaller particle size of 504.4 nm (Figure 3g). The PVP-modified CuO nanoparticles had a quasi-rectangular shape (Figure 3c) and a particle size of 417.9 nm (Figure 3h), which was a 50% size reduction compared to that of the pure CuO nanoparticles. For the PDA-modified CuO nanoparticles, a change in shape to thin, rod-like particles was observed (Figure 3d), and the particle size was substantially reduced to 87.7 nm (Figure 3i). Finally, as shown in Figure 3e, a significant change in shape also occurred for the PVA-modified CuO nanoparticles, which were mainly short, brick-like particles, with a substantially reduced size of 266.5 nm (Figure 3j). The TEM images confirm the successful manipulation of nanoparticle size through the use of different surface-modifying agents. This trend is consistent with the DLS size measurement results obtained, as shown in Table A1.

### 2.2. Surface Potentials of the Polymer-Modified CuO Nanoparticles

Figure 4 shows the surface potentials of the pure and modified CuO nanoparticles in an aqueous environment at pH = 7.0. The zeta potential of the unmodified CuO nanoparticles was −19.68 ± 0.84 mV, indicating a negative surface charge. The surface potential of the PVA-modified CuO was −32.30 ± 0.49 mV, due to the presence of hydroxyl groups in the PVA, generating a greater negative charge on the modified CuO nanoparticle surfaces. In contrast, the surface potentials of the PDA-, PVP-, and PEG-modified CuOs were −1.03 ± 0.85, 7.82 ± 0.75, and 1.32 ± 0.19 mV, respectively, representing positive or near-positive surface charges of the nanoparticles. For the PDA- and PVP-modified CuO nanoparticles, this is due to the presence of amino groups, which result in positively charged modified surfaces [39]. In the case of the PEG-modified CuO nanoparticles, the neutral charge of PEG produces a shielding effect on the negatively charged CuO nanoparticles, leading to a positive surface charge after modification [40].

### 2.3. Antibacterial Activity of the CuO Nanoparticles

An antibacterial experiment was conducted using unmodified and polymer-modified CuO nanoparticles at different concentrations (50, 100, 250, and 500 μg/mL), and the optical density (OD) values (570 nm) at 0, 1, 3, 5, 6, 7, and 8 h were recorded (Figure 5a–e). The bacterial inhibition rates, calculated based on the ratio of the OD value of each experimental group to that of the control group (Equation (1)), are shown in Figure 5f. The results indicated that the bacterial inhibition rates of the pure CuO at concentrations of 50, 100, 250, and 500 μg/mL were 16, 20, 23, and 39%, respectively. Compared to the control group (Figure 5a, green line), the bacterial inhibition effects of the pure CuO increased with increasing antibacterial agent concentration. This preliminarily demonstrated that the inhibition of bacterial growth could be achieved through the addition of CuO, with the rate of inhibition being positively correlated with concentration. When the surface-modified CuO nanoparticles were used as antibacterial agents, the experimental results indicated significantly superior antibacterial effects compared to those of the pure CuO (Figure 5b–e), showing an increase with increased concentrations of modified CuO nanoparticles. The bacterial inhibition rates of CuO-PEG, CuO-PVP, CuO-PDA, and CuO-PVA at the highest dose level (500 μg/mL) were 63, 67, 85, and 57% (Figure 5f), respectively, with CuO-PDA providing the best antibacterial effect. Additionally, the bacterial growth curve for CuO-PDA (Figure 5d) shows a significantly lengthened latent period in *E. coli* growth, resulting from the inability of the bacteria to rapidly enter the exponential growth phase, thereby hindering bacterial growth, and resulting in good bacterial inhibition effects.

Morphological changes in *E. coli* cells subjected to the various antibacterial treatments were observed by using a scanning electron microscope (SEM). As shown in Figure 6, the untreated *E. coli* cells exhibited smooth, intact rod shapes (Figure 6a). By contrast, the *E. coli* cells treated with pure CuO and polymer-modified CuO were severely ruptured 8 h after contact (Figure 6b–f). Based on the results of our previous work, this is due to the release of Cu^2+^ from CuO, which induces the production of reactive oxygen species (ROS) within the bacteria, leading to cell injury and apoptosis [41]. However, the CuO NPs with the largest particle size caused a reduction in stability and dispersion. This led to faster rates of particle deposition and aggregation, which, ultimately, limited the antibacterial activity of the NPS, as shown in Figure A2 and the SEM image in Figure A3. The CuO NPs exhibited an aggregated state in the physiological environments of *E. coli* (pH 7.4 and pH 5), whereas the PDA-CuO NPs possessed excellent stability at pH 7.4. In a low pH 5 environment, the PDA-CuO NPs significantly reduced particle aggregation.

The relationships between antibacterial activity and both surface charge and particle size in three different types of polymer-modified CuO NPs (CuO-PEG, CuO-PVP, and CuO-PVA) were investigated. The negatively charged CuO-PVA demonstrated a smaller particle size than CuO-PEG and CuO-PVP, and the electrostatic repulsion that occurred with the negatively charged surfaces of *E. coli* decreased its antibacterial effects. By contrast, the large-sized CuO-PEG and CuO-PVP NPs were positively charged, which resulted in electrostatic attraction to the negatively charged surfaces of the *E. coli,* and increased the antibacterial abilities of the NPs [42]. Therefore, the surface charge effect dominated the antibacterial activity of polymer-modified CuO NPs.

Notably, the CuO-PDA exhibited accelerated bacteriostatic effects. Unlike other types of polymer-modified CuOs, the CuO-PDA was capable of maintaining a bacteriostatic rate of 85.8 ± 0.2% over a relatively short duration (3 h). This may be attributed to the following three major reasons: (1) As reported in the literature, dopamine, being a catecholamine, is lipophilic at pH 7.4 [43]. Therefore, the exposed catecholamine structures on the surface of CuO-PDA promoted the adherence of the NPs to the lipopolysaccharide-containing outer membrane of the *E. coli* [16,44]. By contrast, the hydroxyl and carbonyl groups exposed on the surfaces of CuO-PEG and CuO-PVP, respectively, were hydrophilic, which reduced their affinity for the *E. coli* surfaces. (2) Although the surfaces of the CuO-PDA NPs were weakly negatively charged, the electrostatic repulsive forces could be overcome by van der Waals forces [45]. (3) The small particle size of the CuO-PDA NPs led to an increase in the area of contact between the bacteria and the NPs, which led to superior antibacterial effects. Lastly, the effects of LB medium on the NPs were explored. The surfaces of all the modified CuO NPs in the LB medium were negatively charged (Figure A4). This was caused by the presence of free ions or polyelectrolytes, which led to a highly negative charge in the LB medium, and was unrelated to bacteriostatic tendency. This result is consistent with the literature report [46]. Therefore, the possibility that the LB medium affects the direct electrostatic attraction of NP materials towards *E. coli* can be eliminated. These results suggest that the interactions of antibacterial NP materials with bacteria may not be solely limited to the electrostatic attraction between charged surfaces. The lipophilic materials may serve a more essential role in achieving greater affinity for the lipopolysaccharide-containing outer membrane of *E. coli* and other Gram-negative bacteria.

## 3. Experimental Methods

### 3.1. Materials

All chemicals used, including cupric nitrate, sodium hydroxide, ethanol, dopamine hydrochloride (PDA) (Sigma, St. Louis, MO, USA), polyethylene glycol (PEG) (Mw: 5000, Sigma, St. Louis, MO, USA), polyvinyl alcohol (PVA) (Mw: 31,000, Fluka bioChemika, Buchs, Switzerland), and polyvinylpyrrolidone (PVP) (Mw: 40,000, Sigma, St. Louis, MO, USA), were of analytical grade and did not require further purification. The chemicals were directly used or dissolved to form sample solutions using ultrapure water with a resistivity of 18.2 MΩ·cm^−1^ at 25 °C from a Milli-Q Type 1 Water Purification System (Merck Millipore, Burlington, MA, USA).

### 3.2. Preparation of CuO Nanoparticles

First, an aqueous solution of 0.5 M cupric nitrate was formulated. Then, aqueous sodium hydroxide solution (5 M, 2.5 mL) was added to 0.75 mL of cupric nitrate solution and continuously mixed for 15 min. Finally, the mixture was added to a Teflon-lined autoclave vessel which was placed in an oven for hydrothermal synthesis at 140 °C for 24 h. After removal of aqueous solution by centrifugation at 5000× *g* rpm for 10 min, the reaction products were successively washed with DI water and 95% alcohol to ensure neutrality and dried at 80 °C to obtain the CuO nanoparticle powder.

### 3.3. Preparation of Polymer-Modified CuO Nanoparticles

Each polymer was added to 0.75 mL of 0.5 M aqueous cupric nitrate solution in a glass reaction flask. The amounts of the various modification agents used were as follows: 750 mg of PEG, 0.6 mL of 1.5% polyvinyl alcohol, 0.75 mL of 2% polyvinylpyrrolidone, and 9 mg of polydopamine. After mixing and reacting for 10 min, 2.5 mL of 5 M aqueous sodium hydroxide solution was added, and the various mixtures were mixed for another 15 min. Each flask was loaded into a Teflon-lined autoclave vessel and placed in an oven for reaction at 140 °C for 24 h. After removal of aqueous solution by centrifugation at 5000× *g* rpm for 10 min, the reaction products were successively washed with DI water and 95% alcohol to ensure neutrality and dried at 80 °C to obtain the different polymer-modified CuO nanoparticles.

### 3.4. Bacterial Culture

A single *E. coli* cell was removed from a primary bacterial plate, soaked in Luria-Bertani (LB) medium, and cultured in a 37 °C incubator with continuous shaking for 16 h to obtain the experimental *E. coli* culture solution. The solution was then diluted to a concentration of cfu count 1.4 × 10^9^ cells/mL using LB medium for the antibacterial experiment.

### 3.5. Antibacterial Experiment

Unmodified and polymer-modified CuO nanoparticles were separately suspended in deionized water at different concentrations (500, 250, 100, and 50 μg/mL). A 1 mL aliquot of each CuO sample suspension was added to an *E. coli* solution (1 mL, cfu count 1.4 × 10^9^ cells), and incubated at 37 °C with continuous shaking for different times (0, 1, 3, 5, 6, 7, and 8 h). Finally, the bacteria suspensions were added to a 96-well culture plate (100 μL/well), and a BioTek Eon microplate spectrophotometer (BioTek Instruments, Winooski, VT, USA) was used to measure their optical density (OD) at 570 nm after 0, 1, 3, 5, 6, 7, and 8 h.

The bacterial inhibition rate of each antibacterial agent was calculated by the following methods: (1) subtracting the initial OD (i.e., OD at 0 h) from the OD at 8 h to eliminate background contributions to the OD values and (2) determining the ratio of the OD in the presence of antibacterial agent to the OD in the absence of antibacterial agent (control group) using Equation (1) [47], as follows:(1)$$\mathrm{Inhibition}\;\mathrm{rate}\;\left(\%\right)=\left[1-\frac{\mathrm{OD}\left(8\;h,\;\mathrm{with}\;\mathrm{CuO}\right)-\mathrm{OD}\left(0\;h,\;\mathrm{with}\;\mathrm{CuO}\right)}{\mathrm{OD}\left(8\;h,\;\mathrm{without}\;\mathrm{CuO}\right)-\mathrm{OD}\left(0\;h,\;\mathrm{without}\;\mathrm{CuO}\right)}\right]\times 100\%$$

### 3.6. Nanoparticle Characteristic Analyses

X-ray diffraction (XRD): An X-ray diffractometer (D2 Phaser; Bruker, Billerica, MA, USA) was used to analyze the powder form unmodified and polymer-modified CuO nanoparticles. Diffraction patterns were obtained using a Cu Kα X-ray source over the range of 5–80°. The average crystallite size (D) was estimated using the Scherrer formula, where the Scherrer constant is 0.9, λ is the x-ray diffraction wavelength (λ = 0.15406 nm), β is the full width at half maximum (FWHM) of plane ($\bar{1}$11) and θ is the Bragg angle in degree.

Fourier-transform infrared (FTIR) spectroscopy: FTIR spectra of the nanoparticle powder samples were obtained using a spectrometer (Spectrum Two; PerkinElmer, Waltham, MA, USA) with a UATR Diamond/ZnSe ATR (single reflection) plate with the following parameters: resolution = 4 cm^−1^; number of scans = 8; scan range = 400–4000 cm^−1^.

Surface charge analysis: Each powder sample was dispersed in ultrapure water, and 1 mL of the dispersed samples was injected into a quartz cuvette and analyzed in a zeta potential analyzer (Beckman Coulter, Brea, CA, USA) at a wavelength of 365 nm using a 30 mW diode laser.

Scanning electron microscopy (SEM): Each powder sample was dispersed in ultrapure water, added to a clean silicon substrate, then coated with gold for 30 s using a sputter coater. Images were obtained using a scanning electron microscope (JSM-6380; JEOL, Tokyo, Japan) with an accelerating voltage of 10 kV.

Transmission electron microscopy (TEM): Each powder sample was dispersed in alcohol solvent, added to a carbon-coated copper grid, and dried at 100 °C. The particle size and morphology of the various samples were observed under a transmission electron microscope (JEM-2100; JEOL, Tokyo, Japan) with an accelerating voltage of 200 kV.

### 3.7. Statistical Analysis

Bacterial inhibition rate data were analyzed using the statistical analysis software SPSS for Windows, version 12.0 (SPSS Inc., Chicago, IL, USA). Data were subject to a one-way ANOVA test and where significant differences among means were found, these were separated by Duncan’s multiple range tests. OriginPro version 8.1 (OriginLab, Northampton, MA, USA) was used for preparing figures.

## 4. Conclusions

We successfully prepared CuO NPs and various polymer-modified CuO NPs by hydrothermal synthesis. Bacterial growth inhibition rates of up to 85.8 ± 0.2% were achieved. The antibacterial mechanisms of CuO were deduced as follows: (1) the positive surface charge of the CuO-PVP enhanced antibacterial activity through electrostatic interactions with the negatively charged surfaces of the *E. coli*. (2) The weakly negatively charged CuO-PDA NPs achieved superior antibacterial effects. This was due to the presence of lipophilic catecholamine structures on the surfaces of the CuO-PDA, which provided good interactive effects with the lipid bilayer in the outer membrane of the *E. coli*, which has hardly been discussed in other literature. In conclusion, we significantly increased the application potential of metal oxides by demonstrating that PDA can be used as a modifying agent to improve their biocompatibility, and that CuO-PDA NPs can feasibly be used as antibacterial agents.

## Figures and Tables

**Figure 1 ijms-22-12913-f001:**
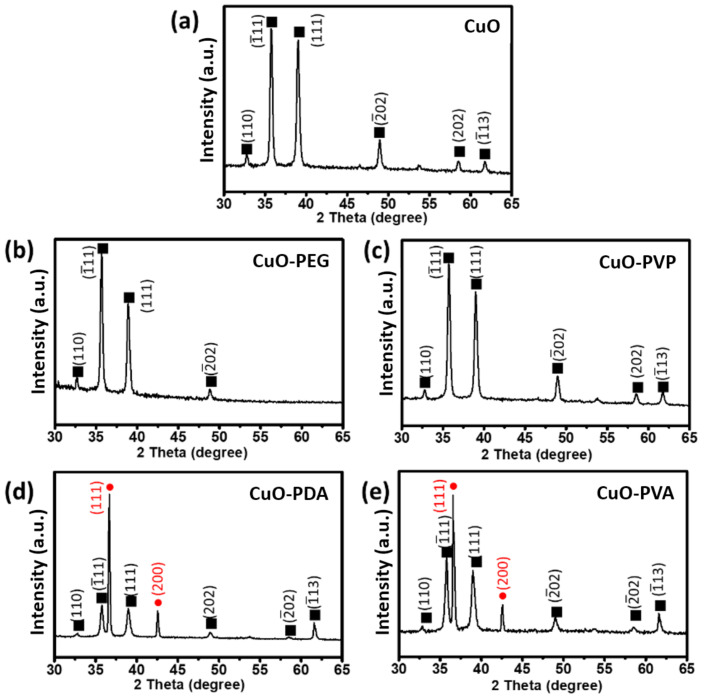
X-ray diffraction (XRD) spectra of (**a**) pure CuO nanoparticles and (**b**–**e**) CuO nanoparticles modified using PEG, PVP, PDA, and PVA, respectively.

**Figure 2 ijms-22-12913-f002:**
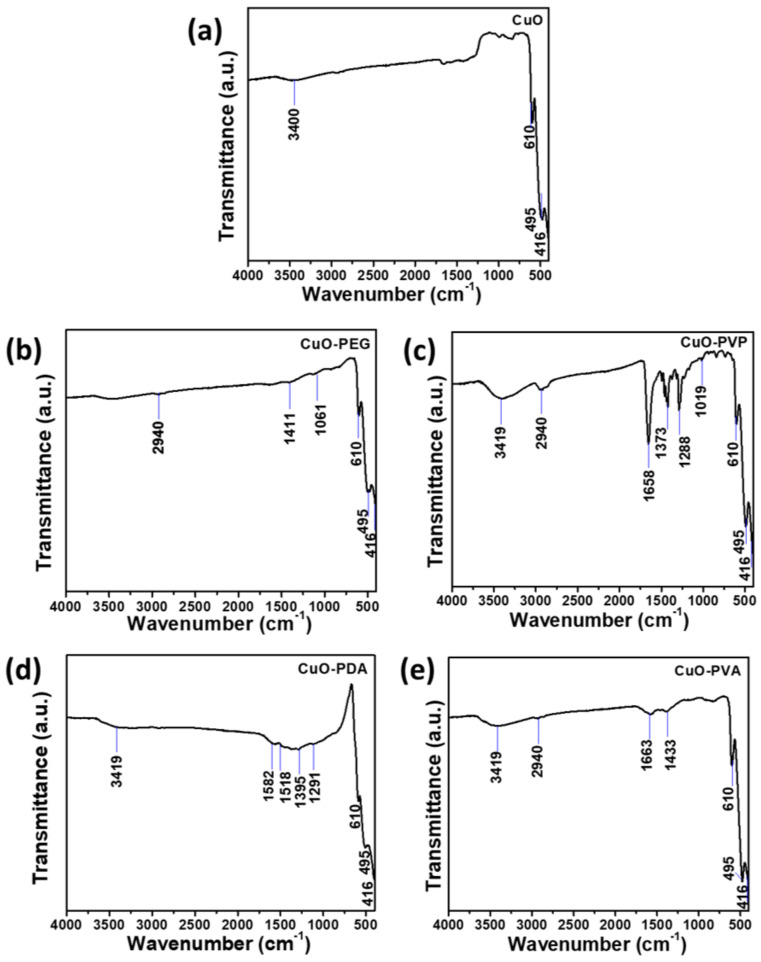
Fourier–transform infrared (FTIR) spectra of (**a**) pure CuO nanoparticles and (**b**–**e**) CuO nanoparticles modified using PEG, PVP, PDA, and PVA, respectively.

**Figure 3 ijms-22-12913-f003:**
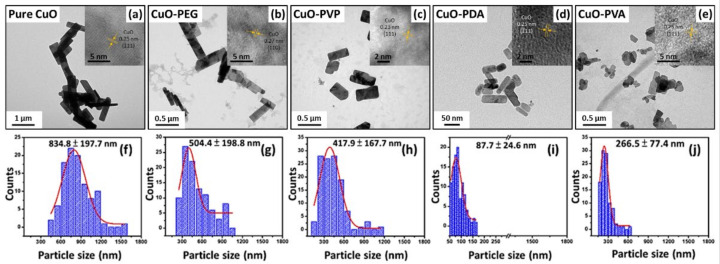
(**a**–**e**): TEM images of the various CuO nanoparticles (pure CuO, CuO-PEG, CuO-PVP, CuO-PDA, and CuO-PVA, respectively) and (**f**–**j**): particle size distribution diagrams corresponding to CuO nanoparticles (**a**–**e**). Insert shows the HRTEM images of the corresponding samples.

**Figure 4 ijms-22-12913-f004:**
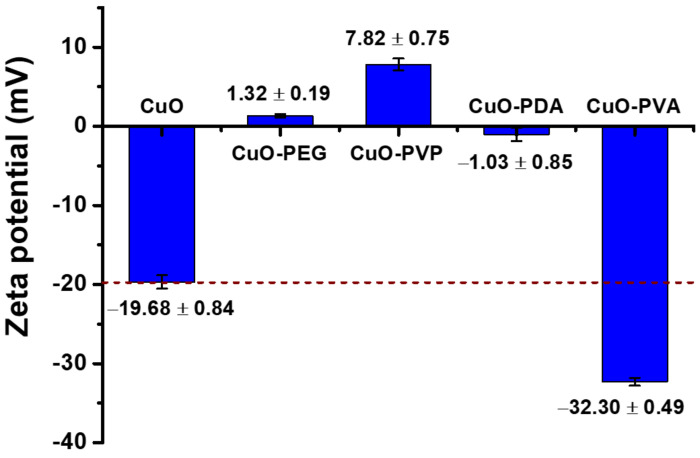
Zeta potentials of pure and modified CuO nanoparticles (detected in water at pH = 7.0).

**Figure 5 ijms-22-12913-f005:**
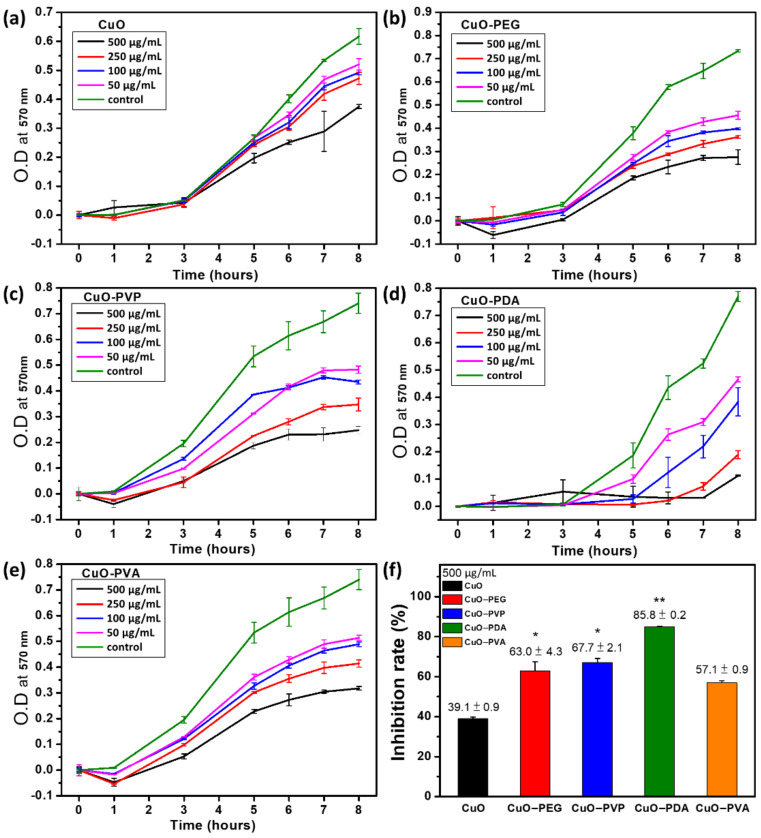
(**a**–**e**): Bacterial growth inhibition curves for pure CuO, CuO-PEG, CuO-PVP, CuO-PDA, and CuO-PVA, respectively, and (**f**): bacterial inhibition rate of (**a**–**e**) at 500 μg/mL concentrations. Statistical analysis was performed using one-way ANOVA followed by Duncan’s test. * *p* < 0.05, and ** *p* < 0.01 versus CuO.

**Figure 6 ijms-22-12913-f006:**
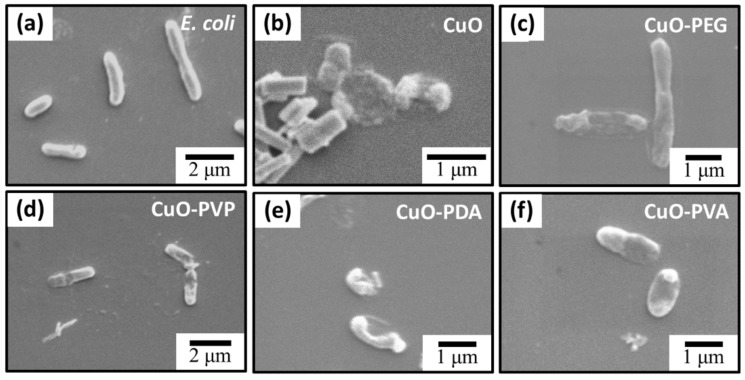
Scanning electron microscope (SEM) images of (**a**) untreated *E. coli* (control) and (**b**–**f**) *E. coli* after 8 h culture with pure CuO, CuO-PEG, CuO-PVP, CuO-PDA, or CuO-PVA, respectively.

## Data Availability

Not applicable.

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
