# Peer review of "Investigation of the Characteristics and Antibacterial Activity of Polymer-Modified Copper Oxide Nanoparticles"

_ijms, 2021, doi:10.3390/ijms222312913_

Round 1

Reviewer 1 Report

The revision of the manuscript has been performed with great care and proper extension. I congratulate the authors for the good work. I don't have further comments to this manuscript since all my previous comments have been addressed and I don't find other issues that deserve modification.

Author Response

Thanks for the reviewer's affirmation.

Sincerely,

Shuchen Hsieh

Reviewer 2 Report

The research paper entitled "Investigation of the characteristics and antibacterial activity of polymer-modified copper oxide nanoparticles (Manuscript ID: ijms-1479775)” was reviewed. This research describes synthesis and characterization of CuO-PDA nanoparticles using various analyses and investigates for antibacterial activity. However, following points are required to address before publication of paper (major revision):

  1. The abstract should be re-written to summarize the work; the abstract should state briefly the purpose of the research, the PRINCIPLE results and MAJOR conclusions. An abstract is often presented separately from the article, so it must be able to stand alone.
  2. In the introduction part, please provide more clear motivation of studying; the author should clarify the uniqueness and advantage of their work compared with other existing works.
  3. Redesign the methods chapter the way so anybody can repeat your procedures, like a recipe
  4. Also add the crystallite size calculation of products from XRD results to have better understanding of the crystallite size.
  5. The authors told that two diffraction signals at 2θ= 36.5 and 42.4° are related to the Cu2O. So, these signals are impurities. Impurities can effect of final efficiency than pure CuO.
  6. Proposed antibacterial mechanism of not clear. Please explain more about mechanism.
  7. Add EDS analysis.
  8. Authors suited on Gram-negative bacteria (such as E. coli). Why not experiment for positive bacteria?
  9. The resolution of TEM analysis should be increased.
  10. The agglomeration and large size of the synthesized nanoparticles can be seen in TEM images. Why?
  11. The author should check the spell and grammar mistakes.
  12. 11. The novelty of this study compared to other studies is not clear.
  13. In introduction section, the authors are not mentioned to various applications of nanoparticles in modern sciences. Need to be improved. Refer to these articles: Journal of Photochemistry and Photobiology B: Biology, 2020, 209, 111949; IET Nanobiotechnology, 2019, 13(6), pp. 560–564; Journal of Rare Earths, 2020, 38(1), pp. 13–20; Surfaces and Interfaces, 2020, 21, 100697; Applied Organometallic Chemistry, 2020, 34(5), e5614; Environmental Technology and Innovation, 2021, 23, 101560; Environmental Technology and Innovation, 2021, 23, 101607.

Round 2

Reviewer 2 Report

Accept

This manuscript is a resubmission of an earlier submission. The following is a list of the peer review reports and author responses from that submission.

Round 1

Reviewer 1 Report

In the present manuscript, the authors Nan-Fu Chen, Yu-Hsiang Liao, Pei-Ying Lin, Wu-Fu Chen, Zhi-Hong Wen and Shuchen Hsieh present the synthesis, characterization and antibacterial properties of the CuO coated with different polymers. The CuO with different properties were prepared with the assistance of hydrothermal synthesis. State of the art in this study is not innovative. The polymers used for the coating/functionalization can be found in many published papers. The introduction part is poorly written, very general without highlighting the punch line of the manuscript. The only reason the polymers used in the study were applied is their charge? What about the antibacterial properties of the applied polymers. The obtained CuO differs in size and surface charge. However, different morphology is also observed. It is known that interplay between size, specific surface area, crystal structure and morphology strongly affect the antibacterial properties. That was not presented in the submitted manuscript. Antibacterial properties were elucidated only on E. coli, no other bacteria were tested? Providing just results without deeper and detailed discussion and connection of the obtained results it is not sufficient to consider this manuscript for the publication. Listed below are comments and suggestions.

  1. Introduction, line 52-56: please check this sentence because it is not clear do you refer to certain class of inorganic NPs or polymers, sugars, etc. used for the coating.
  2. Line 62-64: I disagree with this statement. Please consult the literature for this statement and correct it.
  3. Zeta potential of CuO-PEG and -PDA is around zero and the SD was not presented. According to that, NPs are aggregated. Are the particles stable in water and what happens once when they are introduced into bacterial media? DLS study should be performed. Stability of the CuO NPs is the crucial for the experiments.
  4. There is no info provided about the statistics in the experimental section.

Reviewer 2 Report

I have read with interest the work of Chen et al. regarding the synthesis, characterization and antibacterial activity against E. Coli of CuO nanoparticles and CuO nanoparticles coated with different polymers.

The synthesis and characterization of all the materials employed is well-presented and easy to reproduce in other laboratories, which is the strength of this work. However I have a major consideration about the novelty and significance of the work. Most of the results and conclusions, are, from my point of view, already known. That the modification with polymers can enhance the antibacterial activity of metal oxide nanoparticles and that postive surface charges facilitate the interactions with negatively charged cell membranes is already know. The nanostructures presented are also not new. Thus I would suggest the authors to improve the introduction and discussion section, through relating their work with the scientific literature already available on this topic, and specifically showing which is the differential benefit of the results presented in this work.

As a minor comment, I would also suggest the authors to perform the characterization of the NPs evolution in the media were the antibacterial activity takes place, since the media can modify the characteristics of the original material (e.g. the authors attribute a relevant role to the positive surface charge of some polymer modifited CuO nanoparticles, being CuO-PVP the one bearing the highest surface charge, but being CuO-PDA the one showing higher antibacterial activity. Perhaps the surface charge distribution changes in the biological media)